# Mechanical Properties of Monolayer MoS_2_ with Randomly Distributed Defects

**DOI:** 10.3390/ma13061307

**Published:** 2020-03-13

**Authors:** Mohammed Javeed Akhter, Wacław Kuś, Adam Mrozek, Tadeusz Burczyński

**Affiliations:** 1Institute of Fundamental Technological Research, Polish Academy of Science, Pawińskiego 5B, 02-106 Warszawa, Poland; mjakhter@ippt.pan.pl (M.J.A.); tburczynski@ippt.pan.pl (T.B.); 2Institute of Computational Mechanics, Department of Mechanical Engineering, Faculty of Mechanical Engineering, Silesian University of Technology, Akademicka 2A, 44-100 Gliwice, Poland; 3Centre of Polymer and Carbon Materials, Polish Academy of Science, Marii Skłodowskiej-Curie 34, 41-819 Zabrze, Poland; 4Department of Applied Computer Science and Modeling, Faculty of Metals Engineering and Industrial Computer Science, AGH University of Science and Technology, aleja Adama Mickiewicza 30, 30-059 Kraków, Poland; amrozek@agh.edu.pl

**Keywords:** mono-layer MoS_2_, mechanical properties, molecular statics/dynamics, defects, random distributed defects

## Abstract

The variation of elastic constants stiffness coefficients with respect to different percentage ratios of defects in monolayer molybdenum disulfide (MLMoS_2_) is reported for a particular set of atomistic nanostructural characteristics. The common method suggested is to use conventional defects such as single vacancy or di vacancy, and the recent studies use stone-walled multiple defects for highlighting the differences in the mechanical and electronic properties of 2D materials. Modeling the size influence of monolayer MoS_2_ by generating defects which are randomly distributed for a different percentage from 0% to 25% is considered in the paper. In this work, the geometry of the monolayer MoS_2_ defects modeled as randomized over the domain are taken into account. For simulation, the molecular static method is adopted and study the effect of elastic stiffness parameters of the 2D MoS_2_ material. Our findings reveals that the expansion of defects concentration leads to a decrease in the elastic properties, the sheer decrease in the elastic properties is found at 25%. We also study the diffusion of Molybdenum (Mo) in Sulphur (S) layers of atoms within MoS_2_ with Mo antisite defects. The elastic constants dwindle in the case of antisite defects too, but when compared to pure defects, the reduction was to a smaller extent in monolayer MoS_2_. Nevertheless, the Mo diffusion in sulfur gets to be more and more isotropic with the increase in the defect concentrations and elastic stiffness decreases with antisite defects concentration up to 25%. The distribution of antisite defects plays a vital role in modulating Mo diffusion in sulfur. These results will be helpful and give insights in the design of 2D materials.

## 1. Introduction

The early reports on graphene by Giem [1], specifcally considering two dimensional (2D) materials like transition metal dichalcogenide (MX_2_, M = Mo, W; X = S, Se, Te), or particularly Monolayer MoS_2_ (MLMoS_2_) have lure in the applications of electronics to structural and functional composites [2,3,4,5] owing to its exceptional electrical, optical, and mechanical properties. In structural applications, the most appealing feature of MoS_2_ is the in-sheet elastic stiffness of perfect sp^3^ covalently bonded structures [6,7]. Two dimensional MoS_2_ is tri-layer as opposed to graphene, which is only single-layer, monolayer MoS_2_ having a system of three atomic thickness layers with the transition metal (Mo) atomic layer clogged in the intermediate of two S atom layers [8]. Deji Akinwande [9] reported recent studies related to mechanics, which include interfacial properties and a combination of mechanical and physical properties of 2D materials from both theoretical and experimental aspects. Kai Liu et al. [10] reported the effect of defects on the mechanical properties of graphene and 2D transition metal dichalcogenides, with the elastic properties measured by the experimental nanoindentation technique. Several researchers [11,12,13,14] study the mechanical properties and its effects of defects in MoS_2_ structure, which captures the geometrical aspects of the defects of diverse defect concentrations. It is driven by the simple fact experimentally measured elastic strengths of MoS_2_ significantly corresponds to theoretical expectations for the defect-free structures, and that they exhibit a significant statistical scatter [15]. Theoretical investigations of defected MoS_2_, Li et al., have mainly focused on molecular dynamics (MD) simulations [16], with VMoS_3_ defects using Reactive Empirical Bond Order (REBO) potential. Spirko et al investigated the defects of MoS_2_ structure using DFT method [17]. Wang et al. [18] reported the monolayer MoS_2_ with sulfur vacancy mechanical response under tensile load for electronic and structural properties studied by using first-principles calculations. Besides investigations of the result of single defects, like few studies, Stone–Wales defects and vacancies have addressed the effect of multiple randomly distributed defects. Multiple defects will not just result in a decrease in strength due to microcrack interactions, but might also give rise to a significant scatter in strength related with statistical size effects as larger samples have an enhanced probability to contain weak local configurations. Both experiments [11] and simulation [16] indicate that the strength distribution of MoS_2_ with multiple defects well described by Weibull statistics and exhibit the typical size effects characteristic of weakest link controlled failure. Similar investigations have been conducted in the previous work for MoS_2_ [14,16,18]. 

Samaneh Nasiri [19] investigated the statistical failures analysis in graphene monolayer considering random defects distribution by varying both size and defects concentration. Liu Chu et al. [20] also investigated the effects of random vacancy defects and the tendency due to elastic buckling in single-layer graphene. To this end, chalcogen atoms, i.e., sulfur is the common form of vacancies formed in the TMDs. Shanshan Wang et al. [21] described how sulfur atomic loss is prevalent in MoS_2_ structure and can occur in both top and bottom layers. J.A. Stewart [22] conducted molecular statics nanoindentation on MoS_2_, which leads to phenomena of structural deformation by classical molecular simulations. Electronic structures and elastic properties, in particular, elastic constants of MoS_2_ under pressure conditions, have been examined via first-principle calculations by Li Wei et al. [23]. Lattice defects were introduced by Yan Chen [24] in monolayer MoS_2_ by thermal annealing in a vacuum and by Ar+ ion irradiation. He reported that defects play an influential role in forming the electronic structure by both approaches and the information of defects which can alter the properties of MoS_2_. He introduced lattice defects in monolayer MoS_2_ by both thermal annealings in a vacuum and by Ar^+^ ion irradiation. It has revealed the electronic structure influenced by the lattice defects formed in those ways by both approaches. This information helps to interpret the mechanisms by which the defects alter the properties of MoS_2_. Independent elastic constants by using density functional theory are well described by Nguyen T. Hung [25] as they concentrate on electromechanical properties both 1H and 1T MoS_2_ monolayer as a role of charge doping.

Generally, atomistic simulation packages use the minimum energy configuration to determine the strength of the structure under load, and this is strated by using the conjugate gradient minimization. This minimization gives knowledge about crystal lattice structure in different phases and under different conditions. Molecular static simulations examine the mechanical properties, especially independent elastic constants of monolayer Molybdenum disulfide (MLMoS_2_) containing point defects, like vacancies of Sulphur (S) atoms. In this paper, the defects in the monolayer MoS_2_ are entirely random, and modeling such complex random defects performing many simulations run with molecular statics does capture the elastic constants properties correctly. In this method, each run of the defects simulation with varying defect fraction from 0% to 25%, and for each defect fraction, the elastic properties captured were most of the topological aspects of the monolayer nanosheet. Li et al. and Wang et al. [16,18] proposed the low concentration defects on the topology, whereas we reported for defects varying from low to high concentration to mimic the mono-layer topology in MoS_2_. This method requires multiple defects with the varying defect concentration on the topology generated by using conventional molecular static modeling, namely topology-based atomistic defects caused by using random equilibrium distribution of the domain utilized in this study. Jinhua Hong et al. [26] pointed out that antisite defects with replacing molybdenum as sulfur, which are prevalent point defects in PVD-grown MoS_2_, while the sulfur vacancies are predominant in mechanical exfoliation and CVD specimens in MoS_2_.

The MoS_2_ is also interesting due to the electrical, thermal, and optical properties. The paper [27] presents results of influence of vacancies on electrical properties of MoS_2_, this work focuses on studying the mechanical properties of monolayer MoS_2_ with randomly distributed defects by systematically varying the vacancy concentration, The new results obtained for different levels of defects are presented. The results are shown for different random defect distributions. The results may be used in future works were the mechanical nanomachines and nanosystems based on MoS_2_ sheets optimization are considered. The simulation methodology is detailed in Section 2.1, while the theoretical framework used for defectcted sheet analyzing is defined in Section 2.2. Results are presented in Section 3, and discussion and conclusions are presented in Section 4.

## 2. Materials and Methods 

### 2.1. Methodology of Molecular Simulations

This study investigate the mechanical properties of pristine and defective monolayer MoS_2_ structure including the relaxation strucutre through molecular static simulations and the calculations are performed in atomistic based LAMMPS (Large-Scale Atomic Molecular Massive Parallel Simulator, Sandia National Laboratories, Albuquerque, USA) [28,29], open source package developed by Sandia National Laboratories to model. The heart of the molecular simulations is the inter-atomic potential, which applies to describe the interaction among atoms. From the inter-atomic potential, we can obtain the new properties of any material like theoretical strength, elastic moduli, and Hooke’s law. Stillinger–Weber (SW) potential employs an effective approach to describe the interactions in MoS_2_ by considering all possible interactions between Mo and S [30,31,32].

To understand all its physical properties and know how to control these properties for specific usage, one needs to know the accurate interatomic potential. The potential used in our simulation representing Mo and S atoms developed by Stillinger–Weber (SW) potential of MoS_2,_ which includes all possible interactions Mo and S [30,31] as it is a many-body potential (the potential consists of one-,two- and three-body terms) which perfectly fitted to mono-layer MoS_2_. In this work, we performed molecular static (MS) simulations to generate the elastic constants of the monolayer MoS_2_. The parameters used in the simulation will influence the accuracy of the computed results. Consequently, we used the well parameterized molecular simulation that can describe the variety of bulk material properties. The bond interaction by two-body interaction acts towards the bond deformation while the three-body interaction conducts itself towards the angular rotation. The total potential of a system *φ_tot_* can be written as
(1)$$\varphi_{tot}=\underset{i}{\sum }\underset{i<j}{\sum }Q_{2}\left(r_{ij}\right)+\underset{i}{\sum }\underset{j\neq i}{\sum }\underset{k>j}{\sum }Q_{3}\left(r_{ij},r_{ik},\alpha_{ijk}\right)$$

The two-body interaction potential *Q*_2_ takes the following form.
(2)$$Q_{2}\left(r_{ij}\right)=X_{ij}\left(\frac{Y_{ij}}{r_{ij}{}^{4}}-1\right)e^{\left(\frac{\sigma_{ij}}{r_{ij}-r_{ij}^{max}}\right)}$$

The three-body interaction potential *Q*_3_ in Equation (1) is modeled as
(3)$$Q_{3}\left(r_{ij},r_{ik},\alpha_{ijk}\right)=Z_{ijk}\;e^{\left(\frac{\sigma_{ij}}{r_{ij}-r_{ij}^{max}}+\frac{\sigma_{ik}}{r_{ik}-r_{ik}^{max}}\right)}\times \left(\cos\alpha_{ijk}-\cos\alpha_{0,ijk}\right){}^{2}$$
where exponential function gives a smooth decay of the potential to zero at the cut-off, which is essential to save the energy. *Q*_2_ and *Q*_3_ represents the two body(bond stretching) and three body (bond bending) interactions. The *r_ij_*, *r_ik_* and *α_ijk_* are the pair separations and angle between the separation on atom *i* respectively. The potential parameters are X, Y, Z, σ, along with *r^max^* cutoff radii and equilibrium angles and they rely upon on the atoms interacting with each other, for instance, *Xij* is the parameter for X for the pairwise interaction between atom *i* of category *I* and atom *j* of category *J*.

### 2.2. Crystal Structure

A cubic domain of monolayer MoS_2_ with 1000 atoms was generted using LAMMPS is shown in Figure 1. The crystal orientation was aligned in order that all three principal directions of the crystal align with global coordiante system. The domain has 4.74 × 8.21 × 0.615 nm which is equal to 15a × 15a × 1a, as shown in Figure 1b. The atomistic model in the present work is developed by considering the hexagonal lattice structure of MoS_2_ sheet with the lattice constant of *a* = 3.16 Å and *c* = 6.15 Å. The main objective of this work is to understand the topology of defects by varying size of domain. After the sheets had been relaxed over sufficiently long peiod of time by conjugate gradient optimization till the energy is conserved. The mechanical properties of MoS_2_ are estimated in the next section.

The atomic model consists of 25,000 atoms in the nanosheet for the domain size of 12.95 nm^2^ to 56.11 nm^2^. In this work for each simulation, the initial configuration of the MoS_2_ sheet was prepared, and the random defects were introduced.The periodic boundary conditions in all directions were enforced, and monolayer MoS_2_ sheets with 65 × 65 × 1 lattices are presented in the results section. Point defects (vacancies) are created by randomly removing atoms with probability from the lattice sites. The vacancies or defects are randomly placed in the crystal using a random number generator with uniform distribution. Defect concentrations range from *Vc* = 0.01% to 25% are considered. Hundreds of simulations were performed for each vacancy concentration with different defects locations.

In this work, to see the antisite defects, in particular, the molybdenum (Mo) atoms diffusing in sulfur layers with an assortment of random defect concentrations with different configurations is chosen. There were 5, 56, 120, 296, 571, 867, 1143 and 1427 atoms of molybdenum antisite defects in the MoS_2_ sheet that contains 25,000 atoms, which stand for a defect concentration of 0.1%, 1%, 2%, 5%, 10%, 15%, 20%, and 25%, respectively. Local structures and diffusion dynamics have a significant influence on the interaction of defects for significant defect fraction ratios. Thus, we explore the impact of antisite defects in MoS_2_.

### 2.3. Elastic Constants of Monolayer MoS_2_ Using Molecular Statics (MS)

Molecular statics calculations have been performed using LAMMPS code to generate the elastic constants of MoS_2_ at 0 K. We know that the MoS_2_ sheet consists of a tri-layer, and it experiences strong covalent bonding inward and weak van der Waal’s interaction over the tri-layer due to the polarization effect [22]. The elastic properties of MoS_2_ are determined as the derivative of the stress against the external strain according to Hooke’s law. The generalized Hooke’s law, for the number of independent elastic constants in MoS_2_, is three and can be written as
(4)$$\sigma_{ij}=C_{ij}\varepsilon_{ij}$$

There are three independent elastic constants for MoS_2_, i.e., *C*_11_ is the coefficient of elastic constant relations due to *σ*_11_ to *ε*_11_ similarly for *C*_22_, and *C*_12_. The *C_ij_* values are correlated to the equal volume of the MoS_2_ unit cell. Therefore, the vacuum space has been set up large enough in the z-axis to avoid the interlayer interactions in MoS_2_ monolayer; the *C_ij_* constants then have to rescaled *z* = *t*_0_ to the actual thickness of monolayer MoS_2_. So, we have set *t*_0_ = 6.15 Å, i.e., one half of the out-of-plane lattice constant of bulk MoS_2_. The MoS_2_ structure is fully optimized to its minimum energy by conjugate gradient minimization until the energy is converged. The specific finite lattice distortion of the simulation box leads to a change in energy during convergence, and the respective final elastic constants are obtained [33,34].

The second-order elastic constants for elastic matrix express as: (5)$$C_{ij}=\frac{1}{S_{0}t_{0}}\left(\frac{\partial^{2}E}{d\varepsilon_{i}d\varepsilon_{j}}\right)$$
where *S*_0_ is the area of the sample, *t*_0_ represents the thickness of MoS_2_ monolayer, *E* is the elastic energy, and *ε* is the strain tensor. In polynomial form for 2D materials discussed in [34,35], the elastic energy *E*(*ε*) of MoS_2_ is expressed as: (6)$$E\left(\varepsilon \right)=\frac{1}{2}C_{11}\varepsilon_{xx}^{2}+\frac{1}{2}C_{22}\varepsilon_{yy}^{2}+C_{12}\varepsilon_{xx}\varepsilon_{yy}+2\varepsilon_{66}\varepsilon_{xy}^{2}$$

The *ε_xx_* and *ε_yy_* are the longitudinal strain in *x* and *y* directions and can also be represented as *ε*_1_ and *ε*_2_ respectively in terms of Voight notations, and *ε_xy_* is the applied shear strain in *xy* plane. The MoS_2_ sheet is arranged as zigzag and armchair in the x and y-axis. *ε_ij’s_* and *C_ij’s_* are the corresponding infinitesimal strain tensors and linear elastic constants [34,36]. Born set the benchmark mechanical stability for graphene-like 2D materials, which explains *C*_11_ > 0, *C*_11_ > *C*_12_, and *C*_12_ > 0 and the condition to satisfy for the 2D materials to be isotropic is *C*_11_ ≈ *C*_22_ and *C*_12_. The elastic energy in general for 2D materials for finite distortion is expressed as
(7)$$E\left(\varepsilon \right)=\frac{1}{2}\left(\varepsilon_{1}\varepsilon_{2},2\varepsilon_{6}\right)=\left(\begin{array}{ccc} C_{11} & C_{12} & 0\\ C_{21} & C_{22} & 0\\ 0 & 0 & \frac{C_{11}-C_{12}}{2} \end{array}\right)\left(\begin{array}{c} \varepsilon_{1}\\ \varepsilon_{2}\\ 2\varepsilon_{6} \end{array}\right)$$

After the MoS_2_ sheet is perfectly relaxed or energy is fully converged, the independent elastic constants are extracted for respective strains. 

## 3. Results and Discussion

### 3.1. MoS_2_ Sheet with Pristine and Random Vacancy Defects

Our findings shows that the elastic constants for a MoS_2_ sheet with an infinite system size are *C*_11_ = *C*_22_ = 149.42 N/m, *C*_12_ = 52.29 N/m, which interpret the isotropic nature of the material. One such example of MoS_2_ microstructure with no defects is shown in Figure 2. Open Visualization Tool (OVITO) [37] was used for the visualization of results.

Since the independent elastic constants for a 2D material MoS_2_ are five within the notation employed, the constant elastic tensor is a 3 × 3 symmetric matrix. Due to symmetry, some components vanishes and the elasticity matrix elements will get reduced to three independent elements. The calculated elastic constants of MoS_2_ are shown in Table 1.

All elastic constants *C_ij_* calculated by conjugate gradient minimization using molecular statics simulation in comparison with literature results are given in Table 1. Due to symmetry *C*_11_ ≈ *C*_22_ the obtained elastic constants marginally diverse from the reference data. We can see that *C*_11_ = 149 N/m, which corresponds to a good Young’s modulus of 242 GPa. Bertolazzi et al. [5] reported the elastic stiffness of MoS_2_ monolayer is 180 ± 60 N/m, which corresponds to a good Young’s modulus of 270 ± 100 GPa by the atomic force microscope (AFM) experiment method. The experimental results are higher than our simulation results from this study because in AFM method, tip enforced on the sheet consists of monolayer or multilayer MoS_2_ suspended on the layer incorporate with an array of circular holes are under biaxial tensile stress whereas we have applied uniaxial stress to the monolayer. Li et al. [16] performed MD simulation under the uniaxial test presented that *C*_11_ is found to be 199 GPa for the 1H MoS_2_, our results are while Nguyen T.H. [25] obtained an average young’s modulus of 201 GPa for monolayer MoS_2_. Note the deviation due to the performed DFT calculations, which are derived from the finite difference approach by Thermo-pw code, and we have used latest SW potential which can be used for higher temperatures as well. A comparison of elastic constants from this work are consistent with the experimental and simulation results. Regardless of vacancies, the average value of *C*_12_ and *C*_21_ is used to assess the physical properties of MoS_2_. It is apparent that *C*_12_ ≈ *C*_21_ due to the symmetric stiffness matrix. 

We now describe the effects of modeling monolayer MoS_2_ sheet with randomly distributed defect fraction presented in Figure 3. The geometry optimized average elastic constants for MoS_2_ under different defect fractions are given in Table 2. The elastic constants of monolayer MoS_2_ vs. the defect percentage are illustrated in comparison to the perfect MoS_2_ sheet. The MoS_2_ monolayer sheet is arranged as zig-zag and armchair in *x & y* directions, which denotes the *C*_11_, *C*_12_, and *C*_22_ elastic moduli, respectively. It is clear that chirality slight effect on elastic constants irrespective of defect ratios. The elastic constants *C_ij_* nonetheless started dwindling as the defect fraction piled up from 0% to 25%. Its reduction becomes more expeditiously as the defects grow in the sheet. Piling up the defect fraction and maximum ratio up to 25% results in a considerable decline in the elastic constants, which implies the impact is significant.

The elastic constants of monolayer MoS_2_ nanosheet vs. the defect percentage are presented in Figure 4. The dots denote average values of defects fraction with the fluctuating bar represents the standard deviation that shows maximum and minimum values for hundreds cases with random distributed defects. The result of chirality on the elastic properties of MoS_2_ is negligible, despite the prevailing circumstances of the defect fraction. The elastic constants dwindle faster with the increase in the defect fraction, the maximum reduction of elastic constants is at 25%, more significant than 15%, 10%, and 5%, which implies the influence of defect fraction on the elastic constants is found to be substantial.

To give comprehensive and comparable studies of elastic properties of randomly distributed defects, the elastic constants of MoS_2_ with varying defect ratios have been studied. The elastic constants of this defective MoS_2_ versus defect fraction are shown in Figure 4. For comparison, the elastic constant of pristine MoS_2_ also included in the plots. Figure 4 shows the effect of defects on the elastic constants of MoS_2_ along with the individual independent elastic constants by comparing that of the pristine MoS_2_. It will be hard to conclude the locations of the unperturbed vacancies as they are distributed randomly throughout the layer. We took an interest in determining that these vacancies need to form everywhere in the sheet, irrespective of defect fraction ratio. So, to achieve this, we repeated with different random seeds for a sufficient number of times and estimated the elastic constants for each seed. 

Table 2 displays the average values of the outcomes of the built-in elastic test after repeating the simulation. These results will also motivate us to study the tensile and other properties of MoS_2_ with the defects. With the increase in defect ratio, we observe the difference in the elastic constants nonlinearly as expected. It was found that up to 1% of defects had little impact as it trims down to 2.1% rate of elastic constants when compared to defect-free MoS_2_ and the impact of this vacancy defects on the elastic constants was not that obvious and can be neglected. When the defect fraction surpasses 2%, the elastic constants start trimming down at a rapid rate. Nevertheless, when defect ratios was 2%, 5%, 10%, and 25%, the decrease of the elastic constants was 4.02%, 13.42%, 28.8%, and 56.5%, respectively, compared with MoS_2_ with no defects. This result showed that after exceeding some defect density, the vacancy had a substantial effect and damages the robustness and uniform symmetry of MoS_2_ and has a full impact on the elastic tensile behavior of MoS_2_. It was also found that from Figure 4 as the elastic constants fluctuate within a certain range, and this fluctuation occurs due to the locations of the defects placed randomly and changes its location with each test. These results draw attention towards the foundation in randomly distributed vacancies in MoS_2_ sheets. 

### 3.2. MoS_2_ Sheet with Randomly Diffusing Sulfur to Molybdenum (S→Mo)

The concentration of antisite defects, i.e., sulfur atoms to molybdenum atoms, are also randomly distributed in sulfur layers of the MoS_2_ sheet. Figure 5 shows the diffusion of sulfur to molybdenum as antisite defect for different percentages of sulfur diffused in monolayer MoS_2_ for 25000 atoms nanosheet size. Table 3 shows that the elastic stiffness strengths of 0% to 25% molybdenum doped in sulfur layers in monolayer MoS_2_ for sheet size of 65 Å × 65 Å (25000 atoms) increase by about 0.1%, 2%, 1%, 5%, and an impressive peak of 25% when compared with the MoS_2_ is observed.

The independent elastic constants of 1%, 2%, 5%, 10%, as well as 15% sulphur doped molybdenum in MoS_2_ drops by about 1%, 1.5%, 4%, 8%, 11%, 15%, and 19% in comparison to the pristine MoS_2_. Increasing the percentage of sulfur doping in the MoS_2_ sheet, the elastic properties decrease. Further, we found that the elastic properties due to sulfur vacancy defects with different percentages drop in great detail when compared the elastic properties due to diffusion at the respective percentage as deduce from Table 2 and Table 3. 

Figure 6 shows the effect of antisite defects molybdenum diffusion in sulfur, also dwindle the elastic constants of MoS_2_ for the different defect fractions. The change of fractions were in ranges from 0.1% to 25% antisite defects. We started to pile-up the antisite defects and observed the elastic constants becomes more efficient and started to hinder further with the increase in the diffusion, unlike in case of pure defects where we seen the elastic properties drop dramatically with each defect fraction. The elastic constants *C_ij_* of antisite defect also shows the trend of decrease in nature when compared to defect-free structure, it decreases from 139.69 N/m for 5% defects to 113.83 N/m for 25 % defects when compared to the of pure defects as seen in Table 3. In order to provide the results of impact of the antisite defect in MoS_2_ sheet, again we prepared the random antisite defect models. Hundreds of replications were considered for MoS_2_ with 0% to 25% antisite defects. 

## 4. Conclusions

The elastic constants for MoS_2_ monolayer using molecular statics simulation in great detail were investigated. MoS_2_ is flexible and isotropic for small deformations and the results obtained from this study are compared with the previous literature for defect-free MoS_2_ and progress towards higher defect fractions. The random distribution of defects in the MoS_2_ sheet in addition to antisite defects were also discussed in great detail.

We have seen that the elastic constants of MoS_2_ started dwindling at a rapid rate with the defects pile-up. It started to dwindle at a slow rate of up to 1% of defects. Just as the defects increase to 2% and beyond, its reduction began dramatically. Hence, when the defect percentage reaches 10%, the reduction in elastic constants was as huge as 28.8%. These vacancy defects greatly influenced the elastic behavior of the MoS_2_ lattice. With the increase in defects fraction, the vector sum of displacement affected the geometrical symmetry of the MoS_2_ sheet. Moreover, in this study, we reviewed the possibility of physical properties improvement, and strengthening the elastic stiffness properties due to defects in MoS_2_ was confirmed. 

We also study the elastic constants for Mo as antisite defects, molybdenum diffusion in sulfur layers in the MoS_2_ nanosheet for different defect concentrations ratios ranging from (0.1% to 25%) by using molecular statics simulation. We confirmed that Mo diffusion as the antisite defects indeed decreases the elastic constants in the MoS_2_ nanosheet. Nevertheless, with the increase in defect concentration, Mo diffusion has also shown the decrease in tendency of elastic properties. Mainly, when the antisite defects concentration at 5% to 25%, we see the elastic stiffness decreases less than in comparison to pure defect structures.

## Figures and Tables

**Figure 1 materials-13-01307-f001:**
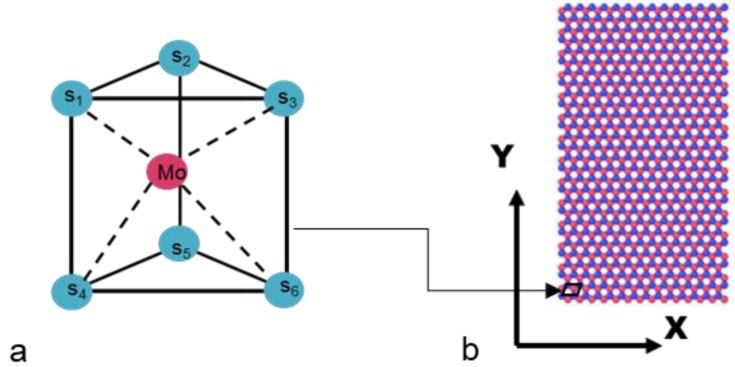
(**a**) Unit cell of hexagonal MoS_2_ (**b**) 15a × 15a × 1a MoS_2_ bulk structure (S = Blue or light; Mo = Purple).

**Figure 2 materials-13-01307-f002:**
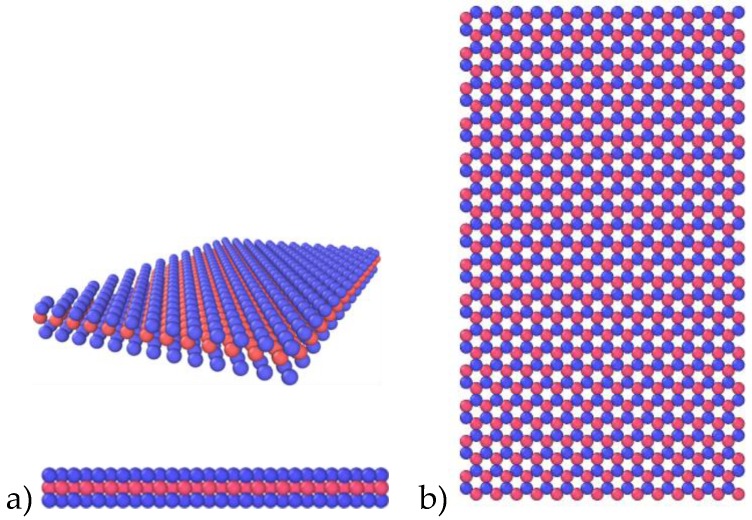
The atomistic model of monolayer MoS_2_ without defects, blue balls represents Sulphur atoms top and bottom layers and red balls represent Molybdenum. The elastic constants for this pristine MoS_2_ are *C*_11_ = *C*_22_ = 149.42 N/m, *C*_12_ = 52.29 N/m. (**a**) Side and isometric view (**b**) top view.

**Figure 3 materials-13-01307-f003:**
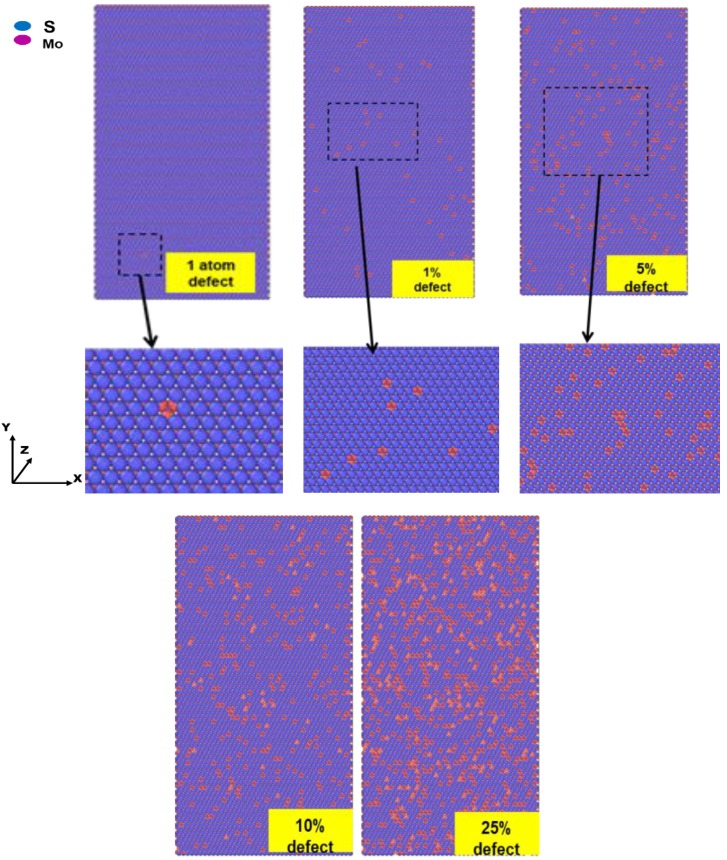
The atomistic model of monolayer MoS_2_ with different percentge of defects.

**Figure 4 materials-13-01307-f004:**
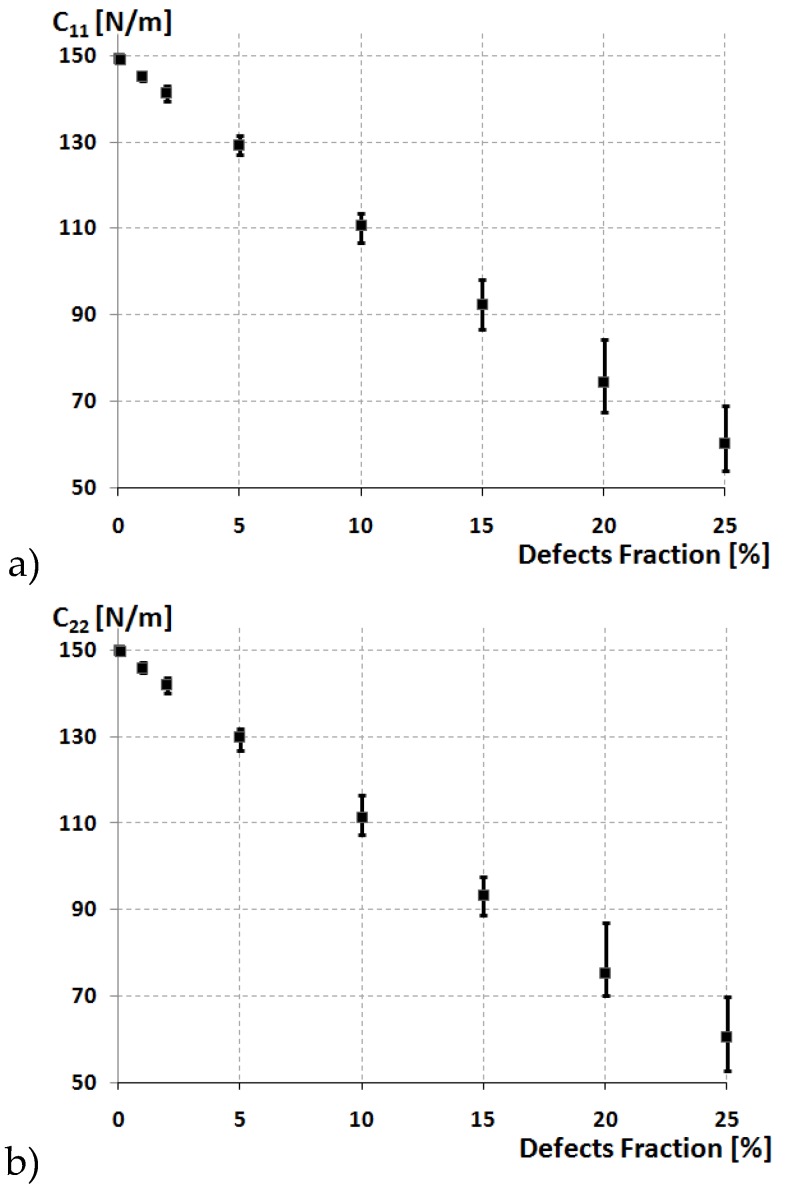
The elastic constants of MoS_2_ as a function of the defect fraction: (**a**) *C*_11_, (**b**) *C*_22_, (**c**) *C*_12_.

**Figure 5 materials-13-01307-f005:**
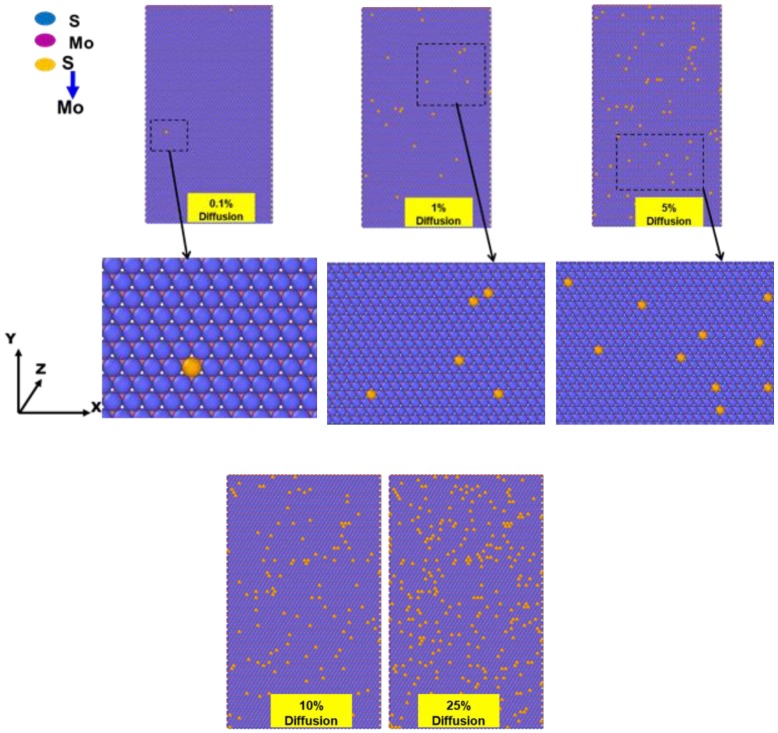
The atomistic model of monolayer MoS_2_ with different percentage of antisite defects.

**Figure 6 materials-13-01307-f006:**
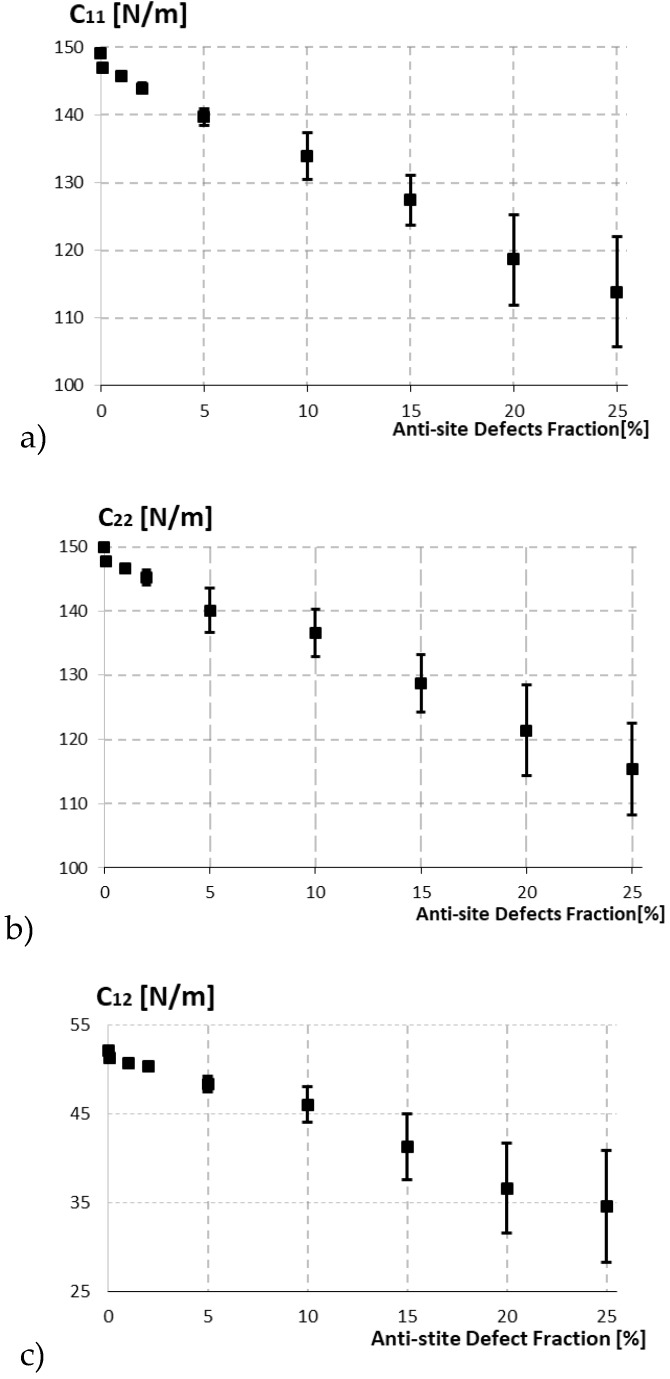
The elastic constants of MoS_2_ as a function antisite defect fraction: (**a**) *C*_11_, (**b**) *C*_22_, (**c**) *C*_12_.

**Table 1 materials-13-01307-t001:** The material properites of MoS_2_.

CODE	*C*_11_ [N/m]	*C*_22_ [N/m]	*C*_12_ [N/m]
This work	149.42	149.42	52.29
Bertolazzi et al. [8]	180 ± 60	180 ± 60	–
Li. M. et al. [19]	148.4	148.4	42.9
Nguyen T.H. et al. [28]	130.4	130.4	26.5

**Table 2 materials-13-01307-t002:** The mechanical parameters for MoS_2_ for different defect fractions along with Standard Deviation (SD).

% of Defects	*C*_11_ [N/m]	SD S(*σ*_*C*11_)	*C*_22_ [N/m]	SD (*σ*_*C*22_)	*C*_12_ [N/m]	SD (*σ*_*C*12_)
Pristine MoS_2_	149.42		149.42		52.29	
Only 1-atom defect	149.11	0.1070	149.04	0.1075	52.23	0.0460
1%	149.01	0.4758	148.85	0.4935	52.15	0.2185
2%	140.70	0.7898	141.51	0.7638	48.52	0.3836
5%	128.89	1.0179	128.89	1.1328	43.25	0.5485
10%	108.30	1.5842	107.94	1.7279	33.70	1.0182
15%	94.21	2.2419	93.47	2.2325	28.75	1.4486
20%	80.99	3.5597	78.46	3.2201	23.16	2.4461
25%	61.15	3.2207	51.52	3.2868	10.67	2.2325

**Table 3 materials-13-01307-t003:** The geometry optimized structural parameters for MoS_2_ for different defect fractions of antisite defects along with Standard Deviation (SD).

% of Diffusion	*C*_11_ [N/m]	SD (*σ*_*C*11_)	*C*_22_ [N/m]	SD (*σ*_*C*22_)	*C*_12_ [N/m]	SD (*σ*_*C*12_)
0% w/o diffusion	149.42		149.42		52.14	
0.1% S→Mo	147.03	0.0996	147.77	0.2300	51.29	0.0658
1% S→Mo	145.82	0.3224	146.60	0.5541	50.77	0.1621
2% S→Mo	143.98	0.4166	145.24	0.5991	50.38	0.1886
5% S→Mo	139.69	0.6203	140.10	0.6698	48.39	0.2296
10% S→Mo	133.91	0.8259	136.61	0.9077	46.04	0.2725
15% S→Mo	127.40	0.9695	128.76	0.9829	41.25	0.3107
20% S→Mo	118.75	0.9553	121.41	0.9556	36.62	0.3009
25% S→Mo	61.15	1.016	51.52	0.9981	10.67	0.3607

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
