# Peer review of "Mechanical Properties of Monolayer MoS2 with Randomly Distributed Defects"

_materials, 2020, doi:10.3390/ma13061307_

Round 1

Reviewer 1 Report

In the present manuscript, the mechanical properties of single trilayer MoS2 with random distributions of vacancy and anti-site defects are investigated by molecular statics simulations.

The work is interesting and the approach looks correct. The manuscript can be considered for publication provided that the following issues are adressed.

  1. The sentence at rows 40-41 'Mono-layer MoS2 is a tri-layer' is someway contradictory. Some variations of the sentence should be considered, such as 'Two-dimensional MoS2 is a tri-layer', or so on.
  2. At rows 74-79 there are some repetitions in citing Chen [27].
  3. The starting of sub-section 3.2 (rows 268-273) is a duplication of sub-section 3.1. The sub-sections 3.1 and 3.2 can be merged in just one sub-section.
  4. However, apart the duplications of points 2. and 3., a full revision of the manuscript's text is recommended. For example, sentence at rows 353-355 'Moreover, in this study, we review the possibility of physical properties improvement and strengthening the elastic stiffness properties due to defects in MoS2 was confirmed' is of difficult understanding of sense.
  5. In the last row of Conclusions a linear decrease of elastic stiffness for Mo diffusion is observed. Really, a linear behaviour of elastic constants vs. defects fraction seems to appear both for vacancy and anti-site defects. This linearity should be better investigated by fitting the points in figures 4 and 6 with a linear equation.

Author Response

1. The sentence at rows 40-41 'Mono-layer MoS2 is a tri-layer' is someway contradictory. Some variations of the sentence should be considered, such as 'Two-dimensional MoS2 is a tri-layer', or so on.

Sentence has been modified.

2. At rows 74-79 there are some repetitions in citing Chen [27].

Sentences have been modified.

3. The starting of sub-section 3.2 (rows 268-273) is a duplication of sub-section 3.1. The sub-sections 3.1 and 3.2 can be merged in just one sub-section.

The subsections has been modified. We would like to separate these two sections due to different modification of atomic structure, in the first one vacancies are present in the structure and in the other one the S-Mo atoms swaps are investigated.

4. However, apart the duplications of points 2. and 3., a full revision of the manuscript's text is recommended. For example, sentence at rows 353-355 'Moreover, in this study, we review the possibility of physical properties improvement and strengthening the elastic stiffness properties due to defects in MoS2 was confirmed' is of difficult understanding of sense.

The manuscript text has been revised and modified.

5. In the last row of Conclusions a linear decrease of elastic stiffness for Mo diffusion is observed. Really, a linear behaviour of elastic constants vs. defects fraction seems to appear both for vacancy and anti-site defects. This linearity should be better investigated by fitting the points in figures 4 and 6 with a linear equation.

This is not linear decrease but “almost linear”, the sentence has been removed from the current version of the manuscript.

Reviewer 2 Report

This manuscript presents simulations to study the effect of elastic stiffness parameters on the atomistic defects in monolayer molybdenum disulfide (MoS2).  The results indicate that elastic stiffness decreases as the defect concentration increases to ~ 25%.  The following comments need to be addressed before consideration for publication.

1. Please elaborate on the model setup in Fig. 1

2. Proper design in the figure is needed (e.g., to efficiently use the space in Fig. 2).  It is also helpful to show the 3D illustration of the atomistic model (over the top/side views).

3. Please elaborate on the discussion of Table 1.  While the discussion simply states "A comparison of elastic constants from the present results is excellent consistent with the experimental and simulation", a large deviation is clearly observed in the table.

4. Please include the standard deviation in the results presented in Tables 2/3.

5. How do the results in Figs. 4/6 compare with those reported in the literature or the experiment?  Validation of the results from this study would be helpful.  

6. Also, given the literature reports about the effect of defects on the mechanical properties of 2D materials, please directly discuss the novelty of this work.

7. The manuscript would significantly benefit from professional editing to help correct typos and grammatical issues.  The equations 1-2 are off from the center as well.

Author Response

1. Please elaborate on the model setup in Fig. 1

The Fig. 1 has been modified to be more clear.

  1. Proper design in the figure is needed (e.g., to efficiently use the space in Fig. 2).  It is also helpful to show the 3D illustration of the atomistic model (over the top/side views).

The iso view of the structure is included in the current version of the paper.

  1. Please elaborate on the discussion of Table 1.  While the discussion simply states "A comparison of elastic constants from the present results is excellent consistent with the experimental and simulation", a large deviation is clearly observed in the table.

The results given in Table 1 show that our results presented in the paper are very close to Li M. et al. results and are within Bertolazzi et al. range of results. The differences of the results are due to the potential parameter used by different authors. The authors of the paper used newest, most reliable SW potential parameters available for MoS2 materials.

  1. Please include the standard deviation in the results presented in Tables 2/3.

The standard deviation has been introduced in current version of the paper.

  1. How do the results in Figs. 4/6 compare with those reported in the literature or the experiment?  Validation of the results from this study would be helpful.  

These results are new and not presented in the other papers. The results for MoS2 without defects were compared (Table 1) with results available in literature.

  1. Also, given the literature reports about the effect of defects on the mechanical properties of 2D materials, please directly discuss the novelty of this work.

The literature reports results for systems with low number of atoms with very low number of defects (one missing atom). The paper is devoted to higher number of defects in the atomic system.

  1. The manuscript would significantly benefit from professional editing to help correct typos and grammatical issues.  The equations 1-2 are off from the center as well.

The manuscript has been modified and improved. The equations have been formatted in the proper way in the current version.

Reviewer 3 Report

This paper presents a simulation study of the mechanical properties, i.e. the elastic constant stiffness coefficients, for monolayer MoS2 in the presence of randomly distributed defects. After a preliminary description of the simulation methodology, results in the case of sulfur vacancies and of  anti-site defects were discussed. The subject of this paper is certainly of interest for the scientific community working on 2D materials. However, I suggest to address the following points to make the manuscript appealing to the broader readership of Materials.

Besides having an impact on the mechanical properties, point and complex defects also play a crucial role in determining the electrical properties of MoS2 and the behavior of devices based on this material. As an example, the unintentional n-type doping of MoS2 and the Fermi level pinning typically observed for metal/MoS2 contacts have been ascribed to defects in the material. See, as references, ACS Nano 8, 2880 (2014); Phys. Rev. B 92, 081307 (2015); J. Phys. Chem. C 123, 5411 (2019); Phys. Status Solidi RRL 14, 1900393 (2020). For the sake of generality, such aspects and relevant references can be mentioned in the Introduction of the paper.  

Author Response

Besides having an impact on the mechanical properties, point and complex defects also play a crucial role in determining the electrical properties of MoS2 and the behavior of devices based on this material. As an example, the unintentional n-type doping of MoS2 and the Fermi level pinning typically observed for metal/MoS2 contacts have been ascribed to defects in the material. See, as references, ACS Nano 8, 2880 (2014); Phys. Rev. B 92, 081307 (2015); J. Phys. Chem. C 123, 5411 (2019); Phys. Status Solidi RRL 14, 1900393 (2020). For the sake of generality, such aspects and relevant references can be mentioned in the Introduction of the paper.  

The importance of electrical, thermal and optical properties of MoS2 have been  indicated in the Introduction section in the current version of the paper.

Reviewer 4 Report

General comments.

  1. The Abstract should be seriously revised. There are sentences without verbs!
  2. In the introduction section the novelty of you work comparing to previous studies should be better underlined. What are the particularities that lead your simulation to results different than of other authors?
  3. The paper needs to be restructured, since results appears in Method section and methods related information to results.
  4. Detailed comments you can find in the attached file.

Author Response

General comments.

1. The Abstract should be seriously revised. There are sentences without verbs!

The Abstract has been modified and corrected.

2. In the introduction section the novelty of you work comparing to previous studies should be better underlined. What are the particularities that lead your simulation to results different than of other authors?

The novelty of the research has been emphasized in the Introduction.

3. The paper needs to be restructured, since results appears in Method section and methods related information to results.

The part of the results has been moved form the Method section into Results.

4. Detailed comments you can find in the attached file.

Thank you for the all comments, they allowed us to improve the paper.

Line 38 The references related to this sentence should be reduced. Also, consider accessing the latest and reliable information in the field.

The references have been modified according to comments.

Line 45 “though a few ab-initio scientific studies have been reported” – consider reformulating.

The sentence has been modified.

Lines 69-70 “The occurrence of a sulfur vacancy in both top and bottom layers of monolayer MoS2 well described by Shanshan Wang et...” – consider reformulating.

The sentence has been reformulated.

Line 101 “This work focuses…” instead of “In this work, we focus…”

The sentence has been modified.

The aim of the paper is to study the “statistical features of defects” or “Mechanical properties of monolayer MoS2 with randomly distributed defects” as the title presents? Please clarify this according to your work.

The goal of the paper is to compute and present mechanical properties, it is clarified in the current version of the paper.

At the end of the Introduction you should refer to: purpose, necessity, novelty, utility of your study.

The additional information has been introduced in the current version of the paper.

Line 110 – verb missing!

Detail the full name, version, standard and producer of “Atomistic based LAMMPS package”.

The sentence has been modified and the exact name of the software has been shown in the current version of the paper.

Line 114&119 “five possible interactions” Are five or three? Please explain.

The sentence has been modified to “all possible interactions”. The term “five” was used because of the number of group of parameters in the potential formulation, and it was not proper here.

Line 119 “many-body” Please check the term.

The many-body potential is common name for the interatomic potential taking into account more than two-body interactions.

Since the study relays only on simulation, please explicitly define the simulation parameters (input variables, initial conditions, constrains, output variables).

A diagram/scheme/drawing representing the interaction can be useful for someone.

The equations 1 and 2 have to be cited and all parameters should be explained.

The initial conditions are given, the potential parameters are taken from literature and citation is provided. The atomic structure is described and the level of defects and S-Mo atoms swaps given. The coefficients in eq. 1 and 2 are described in the current version.

Line 150 What parameters?

The atoms lattice parameters have been explained in the current version of the paper.

The quality of the figures can be improved

The figures have been modified in the current version of the paper.

Line 165 “lots of trials with…” Please specify how many trials and what configurations of defects have been simulated. Can be the defect configurations categorized?

The number of simulations for each level of defects has been introduced into the paper. The few levels of defects concentration were used and for each level simulation were performed hundreds of times with different defects locations (see in Fig. 4)

Lines 168-170 “… the anti-site effect…” This sentence should be rephrased.

The sentence has been rephrased.

Line 174 Verb missing!

The sentence has been modified.

Line 177 There is no standard of determining the elasticity by this method?

The method is described in the paper and it is computed in other papers in similar way.

Line 178 Give details on the coefficients C11, C12, C22.

This part of the paper has been rearranged, the Hooks law (4) has been moved before this line to give idea about coefficients C11,C22,C12.

Line 185 “MoS2 is flexible…”. This sentence is a conclusion and should be removed from Method section.

The sentence has been removed form Method section.

Line 193-194 “flexible energy” ? Are you sure?

It should be the elastic energy. The sentences have been modified.

Line 210-255 This a result! It belongs to the Results section, not Methods.

This part has been moved to Results section.

Line 228-229 Clarify how the elastic constant C11 corresponds to Young modulus? Can you indicate the natural frequency considered in your study?

The MoS2 material with defects is anisotropic material, so one Young modulus cannot be provided in most cases, the elastic constants matrix with Cij coefficients is most convenient way to use in computations. The Young modulus is given for pure structure (isotropic) in the current version of the paper (242GPa). The natural frequencies have been not computed.

What is the reliability of your simulations? Please define all the properties that you used in order to get these results.

The results for pure, without defect material properties, have been compared to the known from literature (also experimental). The well known potential with confirmed parameters has been used.

Line 256 “fluctuating bar” . Is This standard deviation?

The bar shows minimum and maximum value of coefficient. The results were obtained for different random defects locations that lead to different values of material properties.

Line 257 How many??

Why the data scattering Is increasing with the defect percentages?

The high percentage leads to more variations of defects locations, sometimes they may be more clustered and lower material properties, sometimes more scattered and give higher values.

Line 271 “exhibit” . “Present” is not a better term?

The sentence has been modified.

Line 273-279 This sentence belong to the method section.

The sentence has been moved to method section.

Line 280 “exhibit” . “Present” is not a better term?

The sentence has been modified.

There is no statistical analysis here! Only some considerations on the results.

The sentence has been modified. We agree there is no statistical analysis.

Line 298-303 This belongs to Method section!

The sentence has been moved to Method section.

Anti-site or antisite? Be consistent through all the paper.

The term antisite is used in the current version of the paper.

Round 2

Reviewer 1 Report

Some fine/minor spell check is still required.

Reviewer 2 Report

For future reference, it is helpful to include a separate section of "our modifications to the manuscript" for each comment from the reviewer.

Reviewer 4 Report

The authors significantly improve the scientific quality of their work, providing valuable information/discussions that sustain the results. In order to further improve, the paper should be rechecked/revised for language mistakes. There are still sentences without verbs that make the reader confused on what the authors want to express.